# Neochlorogenic Acid Attenuates Hepatic Lipid Accumulation and Inflammation via Regulating miR-34a In Vitro

**DOI:** 10.3390/ijms222313163

**Published:** 2021-12-06

**Authors:** Meng-Hsun Yu, Tung-Wei Hung, Chi-Chih Wang, Sheng-Wen Wu, Tzu-Wei Yang, Ching-Yu Yang, Tsui-Hwa Tseng, Chau-Jong Wang

**Affiliations:** 1Institute of Medicine, Chung Shan Medical University, Taichung 40201, Taiwan; ya780522@gmail.com (M.-H.Y.); apink222@livemail.tw (C.-Y.Y.); 2Department of Health Industry Technology Management, Chung Shan Medical University, Taichung 40201, Taiwan; 3Department of Medicine, Division of Nephrology, Chung Shan Medical University Hospital, Taichung 40201, Taiwan; a6152000@ms34.hinet.net (T.-W.H.); s41111.tw@yahoo.com.tw (S.-W.W.); 4School of Medicine, Chung Shan Medical University, Taichung 40201, Taiwan; bananaudwang@gmail.com (C.-C.W.); joviyoung@gmail.com (T.-W.Y.); 5Division of Gastroenterology and Hepatology, Department of Internal Medicine, Chung Shan Medical University Hospital, Taichung 40201, Taiwan; 6Department of Medical Applied Chemistry, Chung Shan Medical University, Taichung 40201, Taiwan; 7Department of Medical Education, Chung Shan Medical University Hospital, Taichung 40201, Taiwan; 8Department of Medical Research, Chung Shan Medical University Hospital, Taichung 40201, Taiwan

**Keywords:** 5-CQA, NAFLD, fatty acid synthesis, cholesterol synthesis, miR-34a

## Abstract

Neochlorogenic acid (5-Caffeoylquinic acid; 5-CQA), a major phenolic compound isolated from mulberry leaves, possesses anti-oxidative and anti-inflammatory effects. Although it modulates lipid metabolism, the molecular mechanism is unknown. Using an in-vitro model of nonalcoholic fatty liver disease (NAFLD) in which oleic acid (OA) induced lipid accumulation in HepG2 cells, we evaluated the alleviation effect of 5-CQA. We observed that 5-CQA improved OA-induced intracellular lipid accumulation by downregulating sterol regulatory element-binding protein 1 (SREBP1) and fatty acid synthase (FASN) expression, which regulates the fatty acid synthesis, as well as SREBP2 and HMG-CoA reductases (HMG-CoR) expressions, which regulate cholesterol synthesis. Treatment with 5-CQA also increased the expression of fatty acid β-oxidation enzymes. Remarkably, 5-CQA attenuated OA-induced miR-34a expression. A transfection assay with an miR-34a mimic or miR-34a inhibitor revealed that miR-34a suppressed Moreover, Sirtuin 1 (SIRT1) expression and inactivated 5’ adenosine monophosphate-activated protein kinase (AMPK). Our results suggest that 5-CQA alleviates lipid accumulation by downregulating miR-34a, leading to activation of the SIRT1/AMPK pathway.

## 1. Introduction

The liver is the main organ controlling cholesterol and fatty acid metabolism. Excessive lipid accumulation in the liver is associated with several diseases, such as diabetes, atherosclerosis, and NAFLD [1]. NAFLD occurrence may precede the development of more severe liver diseases, including cirrhosis and hepatocellular carcinoma [2]. Therefore, appropriate regulation of hepatic lipid metabolism is crucial to prevent dyslipidemia. The expression of FASN and HMG-CoR, the enzymes essential for the regulation of fatty acid and cholesterol synthesis, is to be considered markers of lipogenesis [3]. Conversely, the expression of carnitine palmitoyl transferase-1 (CPT1) and peroxisome proliferator-activated receptor α (PPARα) is predominantly associated with fatty acid oxidation. In addition, sterol regulatory element-binding proteins (SREBPs) are a family of membrane-bound transcription factors that activate genes encoding enzymes required for cholesterol and unsaturated fatty acid synthesis. SREBP1 activates the transcription of genes required for fatty acid and fatty acid synthesis; SREBP2 modulates the transcription of genes required for cholesterol synthesis [4,5]. It is directly or indirectly modulated by AMPK, an energy-sensing protein complex, to regulate the process of lipid homeostasis [6,7]. Moreover, SIRT1, a nicotinamide adenine dinucleotide-dependent deacetylase, may be required for AMPK activation and the regulation of the transcriptional network of cellular lipid metabolism involved in NAFLD progression [8]. Activation of AMPK in the liver by nutraceutical/pharmaceutical compounds may inhibit fatty acid synthesis and promote fatty acid oxidation [9,10]. Therefore, the SIRT1/AMPK pathway is one of the promising targets in NAFLD prevention.

MicroRNAs are small noncoding RNAs that mediate the post-transcriptional regulation of gene expression by binding to the 3′untranslated regions (3′-UTRs) of the target mRNAs. Dysregulation of miRNAs disrupts the gene regulatory network, leading to the development of metabolic syndrome and related diseases [11,12]. Therefore, modulating miRNA expression using natural products has been highlighted as a mechanism for preventing and treating metabolic diseases or cancers [13,14]. miR-34a has multiple roles in the regulation of cell cycle, apoptosis, and differentiation. In addition, increasing the expression of miR-34a has been indicated to lead to NAFLD development [15,16]. Thus, miR-34a is implied to be a useful target for alleviating lipid-associated metabolic diseases.

Mulberry leaves (*Morus alba* L.), the most common component of a silkworm’s diet, is an edible food used to treat metabolic disorders, such as diabetes, dyslipidemia, fatty liver diseases, and hypertension, in Asia. Our previous study demonstrated the bioactivity of mulberry leaf extract (MLE) against hepatic lipogenesis and inflammation to alleviate liver injury induced by ethanol [17]. MLE has also been reported to prevent obesity-induced NAFLD by regulating adipocytokines, inflammation, and oxidative stress [18,19]. Previous studies have shown that the anti-hepatic steatosis of mulberry leaves involves the regulation of the inflammatory response and autophagy pathway in the liver tissue of HFD-induced obesity mice [20]. Our previous study identified polyphenols in MLE, including rutin, quercetin, chlorogenic acid, and 5-CQA. Among them, 5-CQA was the most abundant polyphenol [17]. Accumulating evidence has demonstrated that 5-CQA exhibits many biological properties, including antibacterial, antioxidative, and anticarcinogenic activities, particularly hypoglycemic and hypolipidemic effects [21,22,23]. A study recently claimed that 5-CQA modulated glucose and lipid metabolism, in vivo, among both healthy individuals and those with genetic metabolism disorders [24,25].

However, the molecular mechanism of the effects of 5-CQA on lipid metabolism is unknown. In the present study, we established an in-vitro model through OA-induced lipid accumulation in HepG2 cells. Our hypothesis is that 5-CQA alleviates hepatic lipid accumulation by downregulating miR-34a, leading to the activation of the SIRT1/AMPK pathway, which modulates the expression of lipogenesis and lipid β-oxidation enzymes as well as lipogenesis-associated transcription factors.

## 2. Results

### 2.1. Effect of 5-CQA and OA on the Viability of HepG2 Cells

NAFLD is also known to be caused by hepatic lipid accumulation and is associated with diabetes and metabolic syndrome [26]. In addition, previous studies showed that OA could induce lipid accumulation in hepatocytes [27]. The MTT assay was performed to evaluate the effect of 5-CQA and OA on the viability of HepG2 cells. As shown in Figure 1A,B, treatment with 50–300 μM 5-CQA for 24 h or 48 h was not cytotoxic to HepG2 cells. In addition, HepG2 cells exhibited no cytotoxicity upon co-treated with 5-CQA (40–120 μM) and OA (300 μM) for 24 h or 48 h (Figure 1C,D). In parallel, cell morphology was not affected by the 5-CQA treatments for 24 h (Figure 1E). Taken together, these results revealed that 5-CQA treatments had no significant effects on the cell viability of HepG2 cells. Therefore, we want to explore the role of 5-CQA in future experiments of OA-induced lipid accumulation of HepG2 cells. Subsequently, we use OA to stimulate the production of lipid accumulation, and subsequent NAFLD condition model (OA cultured HepG2 cells).

### 2.2. Effect of 5-CQA on OA-Induced Lipid Accumulation

Using an Oil Red O staining assay, we detected the intracellular lipid content in HepG2 cells after co-treated with 5-CQA and OA for 24 h to evaluate whether 5-CQA can reduce lipid accumulation. Figure 2A demonstrates the intense red staining in HepG2 cells upon OA administration, whereas co-treatment with 5-CQA decreased the intensity of the color comparatively. Quantitative data demonstrated that 5-CQA significantly reduced OA-induced neutral lipid accumulation in HepG2 cells. Moreover, the Nile red flow cytometry analysis for lipid droplets showed that 5-CQA significantly reduced OA-induced fluorescence (Figure 2B). Taken together, these results revealed that 5-CQA treatments reduced lipid accumulation in a dose-dependent manner.

### 2.3. Effect of 5-CQA on OA-Induced Fatty Acid and Cholesterol Synthesis

To identify the mechanisms involved in the reduction of hepatic lipid accumulation by 5-CQA, we investigated the expression of lipogenic proteins including SREBP, FASN, and HMG-CoR. In Figure 3, 5-CQA significantly reduced the OA-induced expression of SREBP1 and FASN, which are involved in fatty acid synthesis. Furthermore, Figure 4 demonstrates that 5-CQA significantly suppressed OA-induced expression of SREBP2 and HMG-CoR, which are involved in cholesterol synthesis. Taken together, these findings demonstrated that lipogenesis-related proteins were dose-dependently reduced in response to 5-CQA treatments. 

### 2.4. Effect of 5-CQA on OA-Induced Defect of Fatty Acid Oxidation and Inflammation

Several studies have suggested that augmented fatty acid oxidation may protect against steatosis [28,29]. Therefore, we evaluated the effect of 5-CQA on the expression of PPARα and CPT1, which are involved in fatty acid oxidation. As shown in Figure 5, OA treatment reduced the expression of PPARα and CPT1 significantly compared with the control, whereas coadministration with 5-CQA reversed the effects. TNFα has been reported as a mediator of inflammation and plays a critical role in the pathogenesis of NAFLD. We determined the effect of OA on the expression of TNFα and NFκB, which are involved in the inflammatory signal pathway. OA significantly promoted the expression of NFκB and TNFα (Figure 6). Furthermore, co-treated of OA with 5-CQA reduced the OA-induced expression of NFκB and TNFα (Figure 6).

### 2.5. Effect of 5-CQA on the miR-34a/SIRT1/AMPK Pathway

miR-34a has been demonstrated to be a critical regulator of lipid metabolism [16]. In the current study, Figure 7A demonstrates that OA induced miR-34a expression, which was reduced upon co-treatment with 5-CQA. The increase in miR-34a has been reported to be responsible for decreasing the expression of SIRT1, resulting in steatosis [30]. To confirm the contribution of SIRT1 and AMPK to miR-34a regulation, the expression of SIRT1 and phospho-AMPK was determined. As shown in Figure 7B, the expression of SIRT1 and phospho-AMPK was downregulated after transfection with an miR-34a mimic and upregulated after transfection with an miR-34a inhibitor (Figure 7B). The Oil Red O staining assay demonstrated that transfection with the miR-34a inhibitor decreased OA-induced lipid accumulation in HepG2 cells (Figure 7C).

## 3. Discussion

Dysregulation of lipid metabolism leading to excessive lipid accumulation in the liver is a major risk factor for development of life-threatening diseases such as NAFLD. NAFLD is a prevalent chronic disease worldwide and is associated with hypertriglyceridemia, insulin resistance, and metabolic syndrome [26]. The progression of NAFLD may lead to the development of nonalcoholic steatohepatitis, liver fibrosis and cirrhosis, or hepatoma, which have a poor prognosis [2]. In addition to lifestyle adjustment, the major treatment strategy for NAFLD is the development of medicinal or preventive agents to improve lipid homeostasis. In recent years, we found key molecular drivers on treatments were targeted NAFLD, including regulating carbohydrate-responsive element-binding protein (ChREBP) to inhibit obesity and insulin-resistant ob/ob mice to reverse hepatic steatosis. Furthermore, stearoyl-CoA desaturase-1 (SCD1) is one of the key enzymes of lipogenesis, hence the inhibition of SCD1 reduced steatohepatitis [31]. Previous study demonstrated pioglitazone (a thiazolidinedione) is a peroxisome proliferator–activated receptor γ (PPARγ) agonist that improves steatosis, inflammation, and ballooning in patients with NAFLD [32]. F0AT1 (SLC6A19) inhibitor ameliorates the lipotoxicity and dyslipidemia to induce endogenous fibroblast growth factor 21 (FGF21) to treat fatty liver associated diseases [33].

Dietary polyphenols have recently attracted attention for their biological and pharmacological effects. Various studies have reported that natural polyphenols can regulate gene expression through epigenetic machinery to prevent metabolic abnormalities [34]. Luteolin prevents cholesterol synthesis through the reduction of HMGCR expression by suppressing SREBP-2 transcription and post-translational modification in hepatic cells [6]. Naringin treatment can significantly decrease hepatic total cholesterol concentration and hepatic triglyceride concentration. It also can activate AMPK, and lead to changes in the expression of low-density lipoprotein receptor (LDLR), SREBPs and proprotein convertase subtilisin/kexin type 9 (PCSK9), thereby inhibiting obesity in C57BL/6J mice [35]. Epigenetic mechanisms for modulating chromatin accessibility include DNA methylation and histone acetylation, methylation, and phosphorylation. In addition, miRNAs regulate gene expression at the level of translation by degrading mRNA or inhibiting its translation [36]. These mechanisms represent novel targets for polyphenols in the prevention and treatment of diseases. In the present study, 5-CQA, a polyphenol, suppressed OA-upregulated miR-34a expression, implicating that 5-CQA modulated lipid homeostasis through epigenetic machinery. Our previous animal study demonstrated that MLE could prevent NAFLD, and that 5-CQA was the highest phenolic component in MLE [17]. Therefore, 5-CQA was suggested to be a major functional component of MLE. Additionally, the previous study indicated other plants were also found to be rich in 5-CQA, such as caraway, coltsfoot, white mulberry, tarragon, lovage, and green coffee beans, and it was found that 5-CQA improved the proliferation of cancer cells or the exacerbation of the cardiovascular disease [37,38]. Furthermore, 5-CQA induces the activity of antioxidant, and scavenges reactive oxygen species (ROS), which suppresses the expression of inflammation and the inhibition of PPARγ, and then prevents and improves liver steatosis [39]. Thus, 5-CQA can be promising candidates to prevent NAFLD by antioxidant and anti-inflammatory properties.

miRNAs are a class of endogenous noncoding RNAs that can directly target numerous mRNAs, including factors implicated in signal transduction pathways and transcription factors, by modulating gene expression. miRNAs have been implicated to exert crucial functions in various biological processes, including growth, development, metabolic activity, and diseases [40]. Growing evidence has revealed that miRNAs participate in lipid metabolism [41]. Patients with NAFLD have been reported to exhibit an increased expression of miR-34a, which was parallel to the increase in fatty acid accumulation [42]. Conversely, another report demonstrated that reduced miR-34a expression could inhibit lipid accumulation and alleviate hepatocellular steatosis in mouse liver [15]. Consistent with our results, 5-CQA decreased OA-induced lipid accumulation in HepG2 cells, which was associated with the downregulation of miR-34a. In addition, most miRNAs are produced from primary transcripts that are converted into precursor miRNAs (pre-miRNAs) by the RNase III Drosha. Pre-miRNAs are subsequently exported to the cytoplasm, where the RNase III Dicer within the RISC complex cleaves the hairpin of miRNA, resulting in the formation of mature microRNA. Our investigation found that only 5-CQA, an isomer of chlorogenic acid, significantly decreased OA-induced lipid accumulation in HepG2 cells. miRNA expression is a complex process involving multiple enzymes. The stereospecificity of 5-CQA might play a critical role in the expression of miR-34a, which requires further clarification.

The present study was performed using an in-vitro NAFLD model in which OA induced fat deposition and cytokine production in HepG2 cells. We demonstrated that 5-CQA reduced OA-induced intracellular lipid accumulation by suppressing the expression of SREBPs, FASN, and HMG-CoR, which are associated with lipogenesis; 5-CQA also reversed OA-downregulated PPARα and CPT1, which are essential for modulating transportation, uptake, and β-oxidation of fatty acids. In addition, 5-CQA suppressed OA-induced expression of TNF-α and NFκB. Thus, our study results demonstrated that 5-CQA can improve lipid homeostasis.

AMPK is a vital regulator of cellular energy homeostasis. AMPK activation inhibits the expression of acetyl CoA carboxylase and FAS by downregulating SREBPs, thus reducing the synthesis of fatty acids, cholesterols, and triglycerides and promoting fatty acid uptake and β-oxidation [10]. Furthermore, SIRT1, a nicotinamide adenine dinucleotide-dependent deacetylase, can induce mitochondrial biogenesis and suppress the gene transcription involved in adipogenesis [43]. SIRT1 has been observed to be inhibited by miR-34a, and it regulates the activity of AMPK [44]. Polyphenols have been reported to possess therapeutic potential for dyslipidemia by targeting SIRT1/AMPK signaling, which plays an essential role in the regulation of hepatocyte lipid metabolism [45]. Chlorogenic acid has been indicated to activate AMPK, resulting in suppressing hepatic glucose production and fatty acid synthesis [46]. CQA is one of the pivotal polyphenols found in natural food, coffee, and certain plant species, such as mulberry leaves [47,48]. Multiple isomeric forms of CQA, such as 3-CQA, 4-CQA, and 5-CQA, display antioxidant activity [49]. Our study results imply that 5-CQA, an isomer of chlorogenic acid, alleviated NAFLD by modulating the miR-34a/SIRT1/AMPK pathway. In conclusion, our results demonstrated that 5-CQA can downregulate miR-34a, leading to the activation of the SIRT1/AMPK pathway, resulting in the alleviation of hepatic lipid acumination (Figure 8).

## 4. Materials and Methods

### 4.1. Chemicals and Reagents

Dulbecco’s modified Eagle’s medium (DMEM), fetal bovine serum (FBS), penicillin–streptomycin mixed antibiotics, L-glutamine, Dulbecco’s phosphate-buffered saline (PBS), and Trypsin–EDTA were provided by Gibco/BRL (Gaithersburg, MD, USA). Monoclonal antibodies against AMPK (sc-74461), CPTI (sc-20669), SREBP1, SREBP2, HMG-CoA reductase (HMGCR; sc-271595), and PPARα (sc-9000) were obtained from Santa Cruz (Santa Cruz, CA, USA). Cell Signaling Technology Inc (Beverly, MA, USA) provided antibodies against phosphor-AMPK (#2535) and FASN (#3180). The enhanced chemiluminescence kit was purchased from Amersham Life Sciences (Amersham, UK). Polyclonal antibodies against actin, OA, and 5-CQA and other chemical reagents were procured from Sigma Chemical (St. Louis, MO, USA).

### 4.2. Cell Culture

The HepG2 cell line was purchased from the American Type Culture Collection (ATCC, Manassas, VA, USA), and the cells were resuspended in DMEM supplemented with 10% FBS (Gibco BRL Co., Gaithersburg, MD, USA), 1% antibiotics (100 μg/mL streptomycin and 100 U/mL penicillin) (Gibco BRL Co., Gaithersburg, MD, USA), 1% sodium pyruvate (Hyclone, GE Healthcare, Pittsburge, PA, USA), and 2 mM glutamine (Gibco BRL Co., Gaithersburg, MD, USA) at 37 °C in a humidified atmosphere with 5% CO_2_.

### 4.3. MTT Assay

The cells (5 × 10^4^) were plated in 24-well plates and then incubated with OA (300 µM) bound to BSA or with BSA (1 µL /mL) alone as the control, or the cells were co-treated with OA and 5-CQA for 24–48 h. To measure cell viability, 3-(4,5-dimethylthizaol-2-yl)-2,5- diphenyltetrazolium bromide (MTT; 0.2 mg/mL) was added to each well, and incubation continued for 4 h at 37 °C. Thereafter, the culture medium was removed, and the cells were solubilized in DMSO to dissolve the formazan crystals formed. Optical density was measured using a spectrophotometer absorption at 563 nm, and cell viability was calculated.

### 4.4. Oil Red O Staining

HepG2 cells (5 × 10^5^ cells/mL) were co-treated with 5-CQA and OA for 24 h. The cells were fixed with paraformaldehyde (4%) and stained with Oil Red O solution for 10 min at room temperature. After washing thrice, the cells were observed under a light microscope, and a red oil droplet was used in staining the cells to indicate OA-induced lipid accumulation. Finally, to quantify intracellular lipid stained with Oil Red O, isopropanol was added to each sample, and the absorbance was read using a microplate reader absorbance at 500 nm.

### 4.5. Nile Red Flow Cytometry Analysis

To measure cellular neutral lipid droplet accumulation, HepG2 cells were stained using the Nile red method [50]. HepG2 cells were incubated with OA (300 µM) bound to BSA or with BSA (1 µL/mL) alone as the control or were co-treated with OA and 5-CQA for 24 h. After treatment, the cells were harvested with trypsin EDTA, washed three times with ice-cold PBS, and reacted with 1 µg/mL Nile red (Sigma-Aldrich, St. Louis, MO, USA) for 30 min at room temperature. Next, the cells were washed with PBS to remove unbound dye, and they were viewed using flow cytometry (Becton Dickinson Bioscience, San Jose, CA, USA). The cellular lipid content (a minimum of 10,000 cells per sample) was measured using BD biosciences FACscan with CellQuest^TM^ Pro software (Becton Dickinson Bioscience, San Jose, CA, USA).

### 4.6. miRNA Extraction and Real-Time PCR

The total RNA of cells was extracted using the NucleoZOL reagent (Macherey-Nagel, Duren, Germany) according to the manufacturer’s instructions. To quantify the expression of miR-34a, the TaqMan Small RNA Assay kit (Applied Biosystems, Carlsbad, CA, USA) was used to translate RNA to cDNA. According to the manufacturer’s protocol, the Light Cycler Fast Start SYBR Green I Master Mix (Roche Diagnostics, Mannheim, Germany) was used to determine the purity and quantity of miR-34a. The primer sequences used to amplify miR-34a are as following (forward primer: 5′-UGGCAGUGUCUUAGCUGGUUGU-3′; reverse primer: 5′-GUGCAGGGUCCAGGU-3′). In each sample, RNU6B was used as an endogenous (internal) control. After amplification, the expression level of miR-34a was analyzed using Light Cycler software.

### 4.7. Transfection with miR-34a Mimic and Inhibitor

The cells were transfected using a Custom RNA system, the TOOLS Water DNA and RNA extraction kit (Biotools Co., Ltd., New Taipei, Taiwan), according to the manufacturer’s instructions. In brief, 20 pmol miR-34a mimic (5′-UGGCAGUGUCUUAGCUGGUUGU-3′) or miR-34a inhibitor (5′-ACAACCAGCUAAGACACUGCCA-3′) was added to serum-free media and then mixed with 3 μL of T-Pro Non-liposome transfection Reagent II per well for 15 min at room temperature [51,52]. The mixture was added to each culture dish, and the cells were cultured for 24 h at 37 °C in a humidified atmosphere of 95% air with 5% CO_2_.

### 4.8. Western Blot Analysis

After treatment, the medium was removed, and the cells were rinsed with PBS at room temperature. Next, 0.5 mL of cold radio-immunoprecipitation assay (RIPA) buffer (10 µL/mL NP-40, 50 mM Tris base, 1 µL/mL SDS, 5 µL/mL deoxycholic acid, 150 mM NaCl, pH 7.5) with fresh protease inhibitor was added. The cells were scraped, and the lysate was centrifuged at 10,000× *g* for 10 min. The cell lysate (50 µg protein/sample) was mixed with an equal volume of RIPA buffer and then boiled for 10 min, followed by analysis using sodium dodecyl sulfate polyacrylamide gel electrophoresis (SDS-PAGE). The protein was transferred from the gel to the nitrocellulose membrane (Millipore, Bedford, MA, USA) by using an electro-blotting apparatus. The membrane was further incubated with the indicated primary antibody, followed by incubation with a secondary antibody conjugated with horseradish peroxidase. The proteins were then visualized with Enhanced chemiluminescence (ECL) blotting detection reagents (Amersham Biosciences, Mountain View, CA, USA), and densitometric analysis was performed using the Fuji LAS-3000 imaging system (FUJFILM, Tokyo, Japan).

### 4.9. Statistical Analysis

All statistical analyses were performed using GraphPad Prism 7.0 (GraphPad Software, San Diego, CA, USA), and descriptive statistics are presented as mean ± standard deviation of three independent experiments. Between-group differences were analyzed using the Student *t*-test. Moreover, *p* < 0.05 was considered statistically significant.

## Figures and Tables

**Figure 1 ijms-22-13163-f001:**
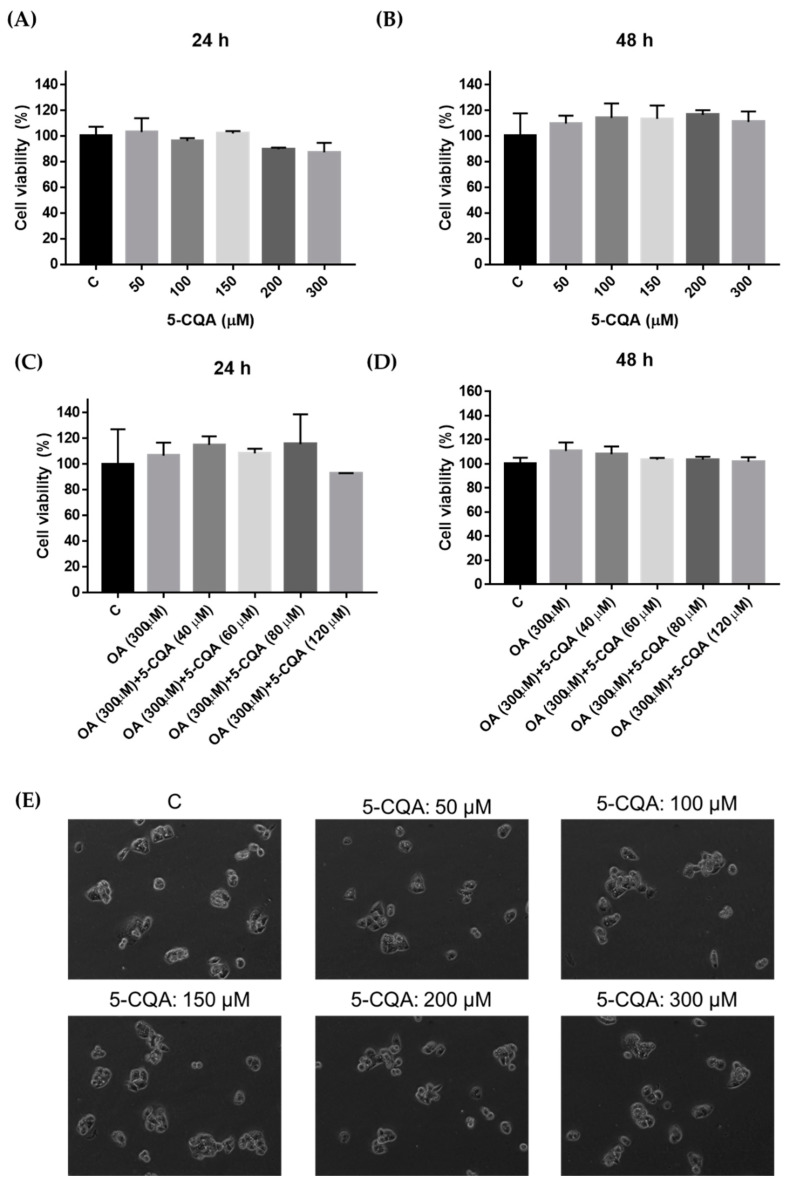
Cytotoxicity of HepG2 cells after treatment with 5-CQA. (**A**) Cultured HepG2 cells were treated with various concentrations of 5-CQA for 24 h and analyzed using the MTT assay. (**B**) Cultured HepG2 cells were treated with various concentrations of 5-CQA for 48 h and analyzed using the MTT assay. (**C**) Cultured HepG2 cells were co-treated with various concentrations of 5-CQA with OA (300 μM) for 24 h and analyzed using the MTT assay. (**D**) Cultured HepG2 cells were co-treated with various concentrations of 5-CQA with OA (300 μM) for 48 h and were then analyzed using the MTT assay. (**E**) Cells were treated with indicated concentrations of 5-CQA for 24 h and then the cell morphology was monitored by phase-contrast microscopy. The data are expressed as mean ± SD from three samples for each group.

**Figure 2 ijms-22-13163-f002:**
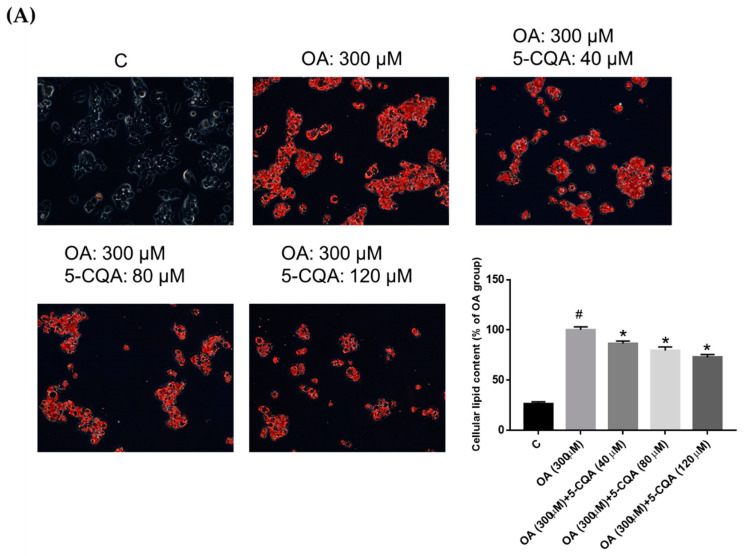
5-CQA attenuated oleic acid (OA)-induced lipid accumulation in HepG2 cells. (**A**) Cultured cells demonstrated lipid accumulation induced by OA (300 μM) for 24 h. Cells were co-treated with OA and indicated concentrations of 5-CQA (40, 80, or 120 μM) for 24 h. Cellular neutral lipid was stained with Oil Red O and the representative photo was taken. Quantitation was determined using colorimetric analysis as described in the text. (**B**) Lipid accumulation was determined by the Nile red flow cytometry analysis. The cellular lipid content (a minimum of 10,000 cells per sample) was measured using BD biosciences FACscan with CellQuestTM Pro software. The data are expressed as mean ± SD from three independent samples for each group. # *p* < 0.05, in relation to the control group. * *p* < 0.05, in relation to the OA-induced group.

**Figure 3 ijms-22-13163-f003:**
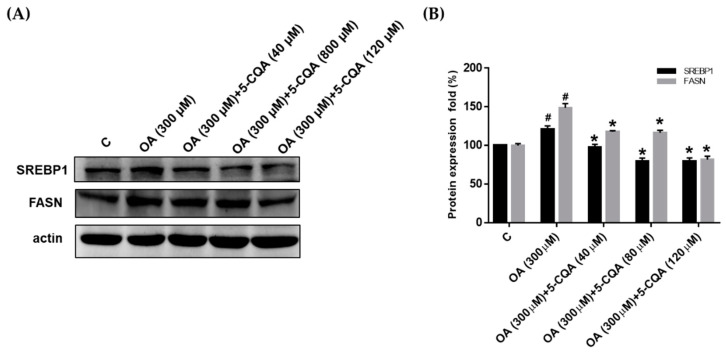
5-CQA decreased the expression of enzymes involved in fatty acid synthesis. (**A**) The cells were treated with OA with or without 5-CQA (40, 80, or 120 μM) for 24 h. Protein expression was detected by Western blot analysis against SREBP1 and FASN antibodies, with anti-actin as the internal control. (**B**) The data are presented as mean ± SD from three samples for each group. # *p* < 0.05, in relation to control group. * *p* < 0.05, in relation to OA-induced group.

**Figure 4 ijms-22-13163-f004:**
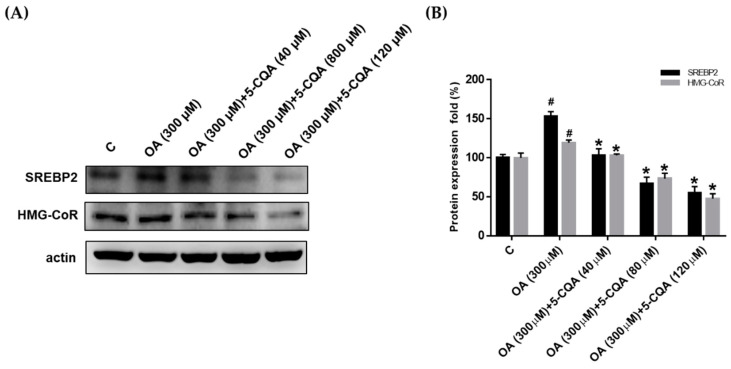
5-CQA decreased the expression of enzymes involved in cholesterol synthesis. (**A**) Cultured cells were treated with OA with or without 5-CQA (40, 80, or 120 μM) for 24 h. Protein expression was detected using Western blot analysis against SREBP2 and HMG-CoR antibodies, with anti-actin as the internal control. (**B**) The data are represented as mean ± SD from three samples for each group. # *p* < 0.05, in relation to the control group. * *p* < 0.05, in relation to the OA-induced group.

**Figure 5 ijms-22-13163-f005:**
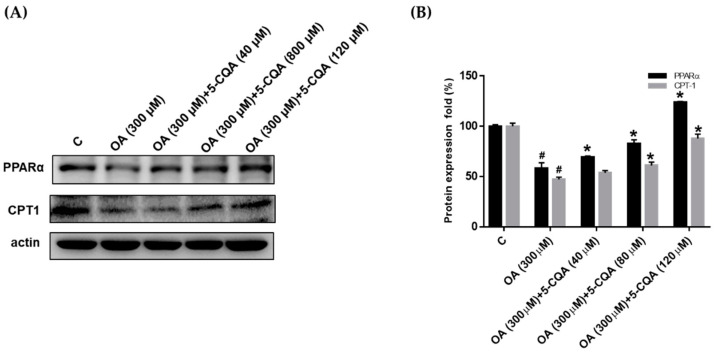
5-CQA increased the expression of enzymes involved in fatty acid oxidation. (**A**) Cultured cells were treated with OA with or without 5-CQA (40, 80, or 120 μM) for 24 h. Protein expression was detected using Western blot analysis against PPARα and CPT1 antibody, with anti-actin as the internal control. (**B**) The data are represented as mean ± SD from three samples for each group. # *p* < 0.05, in relation to control group. * *p* < 0.05, in relation to OA-induced group.

**Figure 6 ijms-22-13163-f006:**
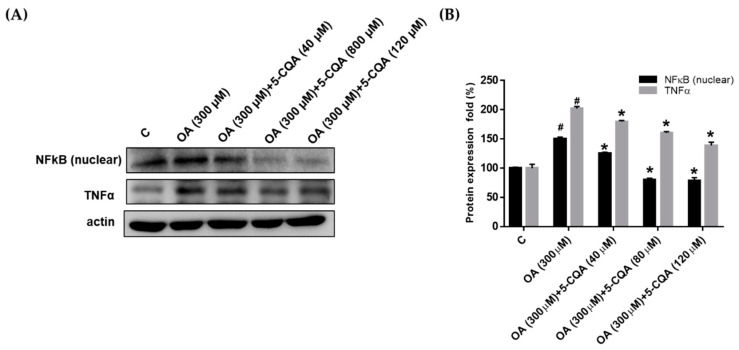
5-CQA decreased the expression of inflammatory mediators. (**A**) Cultured cells were treated with OA with or without 5-CQA (40, 80, or 120 μM) for 24 h. Protein expression was detected using Western blot analysis against NFκB (nuclear) and TNFα antibody, with anti-actin as the internal control. (**B**) The data are represented as mean ± SD from three samples for each group. # *p* < 0.05, in relation to control group. * *p* < 0.05, in relation to OA-induced group.

**Figure 7 ijms-22-13163-f007:**
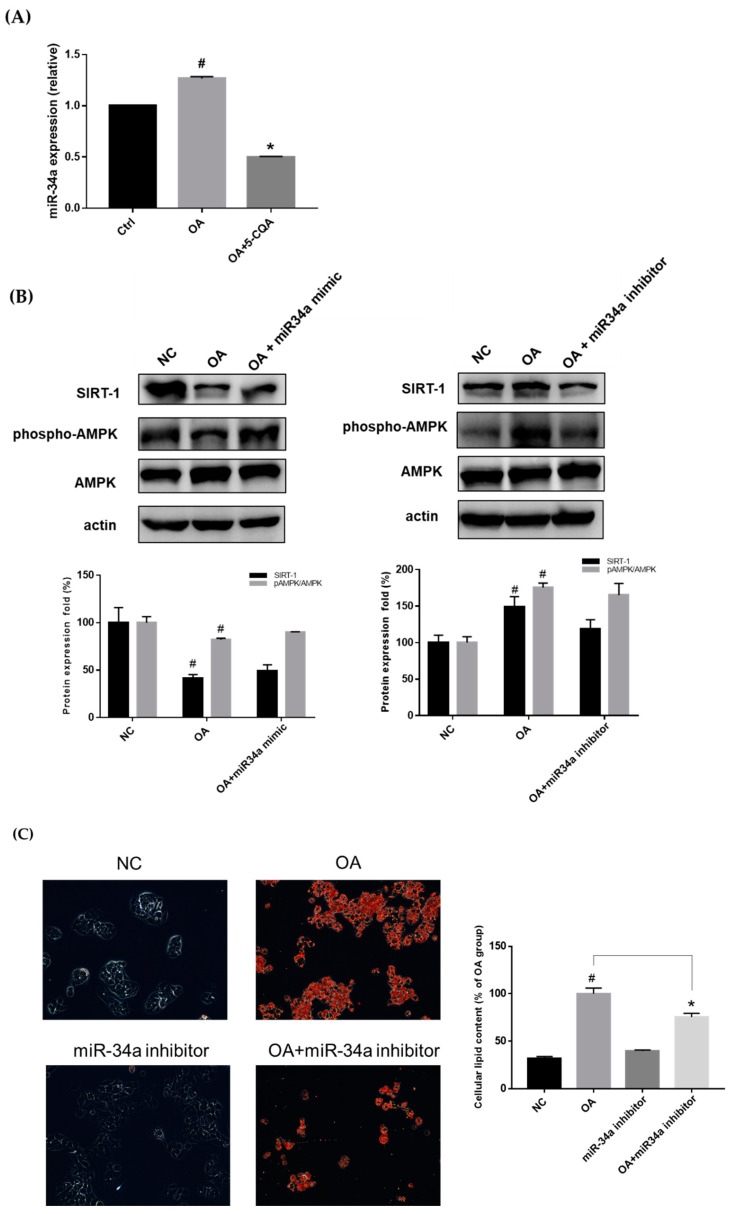
The effect of 5-CQA on miR-34a expression and the effect of miR-34a on lipid accumulation. (**A**) The cells were co-treated with OA and 5-CQA for 24 h. The level of miR-34a was analyzed as described in the text. Treatment with 5-CQA decreased the expression of miR-34a. The results are presented as mean ± SD from three independent experiments for each group. # *p* < 0.05, in relation to the control group. * *p* < 0.05, in relation to the OA-induced group. Ctrl, control; OA, oleic acid. (**B**) After transfection with miR-34a mimic or inhibitor, the total cell lysate (protein) was analyzed using immunoblotting against SIRT1, phosphor-AMPK, AMPK, and actin antibody. The data are presented as mean ± SD. # *p* < 0.05, in relation to the negative control group. (**C**) After miR-34a inhibitor transfection, cells were treated with or without OA for 24 h. The intracellular lipid was stained with Oil Red O and the representative photo was taken. The quantification was determined using colorimetric assay as described in the text. The data are presented as the mean ± SD of three samples of each group. # *p* < 0.05, in relation to the negative control group. * *p* < 0.05, in relation to the OA-treated group.

**Figure 8 ijms-22-13163-f008:**
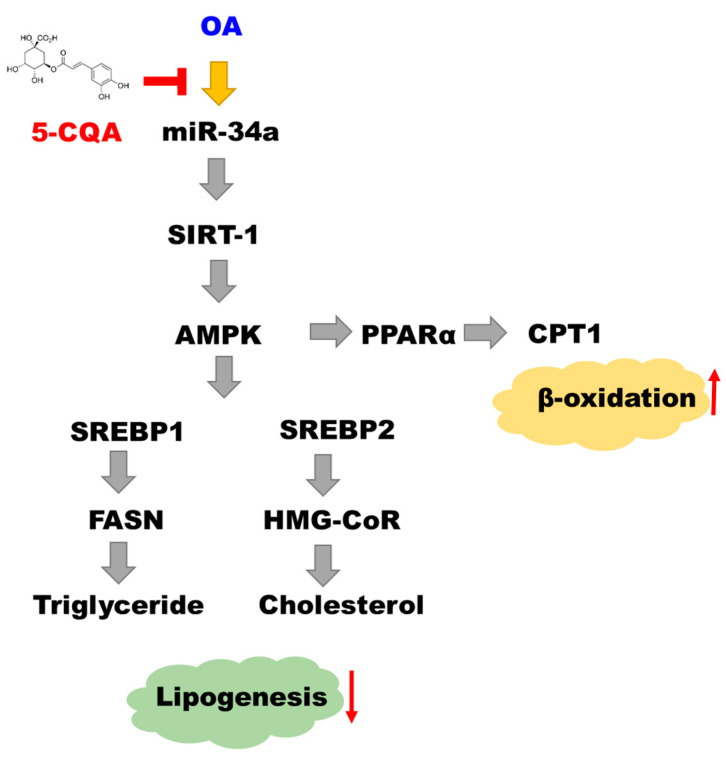
The mechanism by which 5-CQA alleviates lipid accumulation.

## Data Availability

Not applicable.

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
