# Peer review of "Neochlorogenic Acid Attenuates Hepatic Lipid Accumulation and Inflammation via Regulating miR-34a In Vitro"

_ijms, 2021, doi:10.3390/ijms222313163_

Round 1
Reviewer 1 Report
General Comments:
This paper purports to show that the treatment with mulberry leaf metabolite 5-CQA can inhibit/prevent the oleic acid induced accumulation of fatty acids in HepG2 cells by inhibiting/reducing the expression of microRNA miR-34a, which in turn activates the SIRT1/AMPK pathway, resulting in the reduced accumulation of lipids in the cells. Whilst in principle these findings are interesting there are several controls experiments missing which would validate the main finding. I have requested these control experiments below in the comments as well as requesting a major revision of the papers referencing.
Minor Point/edits
- The referencing of the manuscript needs some work. In general, there are general reviews referenced where specific primary research articles should be referenced. In addition, there are passages where no references are given at all. I will detail the significant problems below under minor points for specific sections but, in general, the authors need to add substantial primary research papers to the introduction and discussion where they have simply cited general, and often not very appropriate, reviews. Please take the time reference accurately and comprehensively - if several research papers contributed to the information being provides then please reference them all. This includes, but is not exhaustive of:
- The introduction sentence ending '...is to be considered markers for lipogenesis.' needs primary research references.
- ref 2, 3 - replace with the primary research papers. This is particularly relevant for ref 3, where it is indicated that SREBP2 may be 'directly or indirectly' modulated by AMPK - an expression which indicates multiple research findings in this field. Moreover, what is directly and specifically known concerning how AMPK/SIRT1 regulate fatty acid synthesis via the actions of SREBP2; in section 2.5 you seem to indicate the literature shows depression of SIRT1 cause steatosis, which would imply that SIRT1/AMPK depress SREBP1/2-mediate biosynthesis of fatty acids/sterols. Is this the case? Please be more specific and direct (with the correct references) in the information you provide so that the reader can follow exactly what is known in the current research.
- ref 5 - although it is fine to use a generic review for this point, surely there are more up-to-date and, therefore, accurate and comprehensive reviews that the author's can direct the reader to.
- ref 6 - requires the primary research paper(s) which established that increased miR-34a expression facilitates NAFLD development.
- ref 9 - please provide the primary research references which cumulatively demonstrate the biological properties, not simply a review. These observed effects are central to 5-CQA action and, therefore, are central to your introduction.
- ref 10 - is a review but you directly refer in the sentence to a specific research finding(s) - what are they? Replace the review with the primary research papers
- ref 11 is confusing; you state the 'current paper' establishes the model for OA accumulation in HepG2 cells but you provide this reference. Is the establishment of a cell culture model in HepG2 cells for NAFLD part of the novelty of this paper or not? And if not where is the primary research for the model used as the paper referenced (again) seems to be a review.
- The following publications also provide evidence for the effect of mulberry leaf extract on NAFLD or hepatic lipogenesis in addition to your cited reference 8. They should, therefore, be also referenced and are highly relevant to a complete understanding of this research field for the non-expert: PMID 26463593, PMID: 29023387, PMID: 29970566, PMID: 30643537, PMID: 32948162. The author's should also extend the information of provided in these research papers into the final paragraph of their introduction to give a more complete and correct picture of the state of the research field as a rationale for this particualr research.
- ref 12 - is a review yet you have stated that there are 'several studies' suggested augmentation on fatty acid oxidation may protect against steatosis. Please provide these primary research references.
- section 2.5 refs: reference 13 seems to be incorrect, or at least not the only citation for establishing the reported link between miR-34 and SIRT1. I found many papers establishing a link between the two via a simple PubMed search. Please ensure that the reference provide is the first and only paper reporting the link between these 2 in the context of developing steatosis. For ref 6, please replace with primary research papers as stated above.
- Ref 14: remove the review or at least include the primary research paper(s) for this information. This information should also perhaps be provided in the introduction not in section 2.5.
- There are several recent proposed and substantiated treatments for NAFLD and liver steatosis that have been proposed or demonstrated in the literature that are worth pointing out to the general reader and are relevant to this research for giving context to the field. Not the least because the directly targeting of lipid accumulation or the activation of PPARα to combat NAFLD are not the only proposed treatment methods. I would suggest the authors quote the following publications to illustrate these points: PMID: 34667947 and PMID: 34710482 (for excellent up-to-date overviews of current treatments and proposed treatments) and which list comprehensively these current treatment options.
2. The last sentence of the introduction is repeated.
3. Results 2.1: MMT assay - what is this? Provide a expansion of the acronym and add it the heading for section 4.3 of the methods so that the reader can more easily locate which assay is being used for figure 1.
4. On a more significant point on the MTT assay: you give indications in the intro and the abstract that you are developing and providing a new method to measure fatty acid build-up of OA for this paper, yet here in the results and method section 4.3 it seems that this is not a 'model' at all but just an established assay to measure cell viability (MTT assay)? In other words there is a dichotomy between how the paper is presented in the intro and how the results/method are reported. Please clarify whether you are establishing a model here and what you propose that model to be - is it , for example, that you propose this OA accumulation is a cell culture model for NAFLD or simply lipid accumulation in HepG2 cells, in which case it is not a model at all but just an assay. If you are providing this HepG2 OA-incubated system a a model of a disease condition (NAFLD) what evidence do you have, and what particular cellular markers of NAFLD do you use, to validate it as a realistic model with which to study this condition? These points need some clarification and/or validation.
5. Results 2.1: You also need a conclusion sentence to this short section to give context to the result with the wider aim: obviously it shows that 5-CQA nor 5-CQA+OA are cytotoxic, but why is this important to establish? I'm guessing so that you can the cells and the treatment for further experiments but please make this explicit.
6. Please change the x-axis for Figs. 1C and 1D - as they currently read they give the (misleading) impression) that the concentration numbers provided on the axis are the combined concentration of OA and 5-CQA. Clearly from the figure legend it rather that the OA concentration is constant at 300 uM.
7. It would nice to provide microscopic images of the cells treated as in Fig. 1 so that the reader can observe that there are no morphological changes in the cells as well as no changes in their viability.
8. The staining of OA is quite difficult to see in the images provided - could the authors make the Oil Red O staining brighter in the published images?
9. What is the Control in Fig. 2A, clearly it is Oil red O detection in non-treated cells but why then can we see the outline of HepG2 cells when this cannot be seen in other panels of this figure? In other words, what is the white that can be seen in the control?
10. So you co-incubated 5-CQA with OA in Fig 2. - did you attempt the sequential addition of 5-CQA to OA-treated cells? i.e. did you incubate the combined conditions with OA for 24hrs and then add 5-CQA after? I would be interested to see if the effect is reversible following the accumulation of OA 300 uM as was done in the first experiment panel as opposed to preventative, which is what the experiment you have shown implies
11. Fig. 2B is not well described. I take it the various flow cytometry panels show a shift in the distribution back toward the ctrl panel as the 5-CQA concentration is increased? Is it not possible to overlay the distributions of all treatments on the same panel so they can be compared more easily?
12. the flow cytometry method need a more detailed description than the single sentence provided in the methods section 4.5.
13. At the start of section 2.3 you write 'To identify the mechanisms involved in the inhibition of hepatic lipid accumulation by 5-CQA...' However, you have not exactly demonstrated that 5-CQA inhibits lipid accumulation for the reason I stated in point 10; you cannot, based on Fig. 2, state whether the reduction in OA in the HepG2 cells is being 'inhibited' or 'reversed' as you have only tested this effect but adding both OA and 5-CQA simultaneously. The control experiment where 5-CQA is added after incubation in OA for 24hrs needs to be conducted.
14. Please label the individual panels of Figs. 3, 4, 5, 6 as A/B to make it easier for the reader to follow.
15. Section 2.3 results are fine but please add more detail into you analysis in the text i.e. be specific - which time-point and to what extent? This assists the reader in focusing on the relevant result and also emphasizes the important results in each figure.
16. Figure 7C: similar to point 13 above, what happens if you incubate the cells in OA prior to transfection with OA instead of what you had done which is to transfect with miR-34 first and then add OA? In other words, is miR-34a inhibition preventing or reversing fatty acid accumulation?
17. Section 2.6 is incorrectly labeled section 2.5.
18. section 2.6 is a discussion point and a model to summarise you results - please move it to the discussion and use this model figure to guide your discussion.
19. Figure 7A is a key result for your conclusion that links the effect of miR-34a (already known as far as I can see in the literature) to the action of 5-CQA. I would like to see an additional experiment to further confirm that the reduction in miR-34a is really due to the specific effect of 5-CQA. Please provide an additional control with the cells incubated as in fig. 7A but with 5-CQA solvent i.e. vehicle control. In additional could you provide the same experiment but with either 3- or 4- CQA treatment to see if the effect is specific to 5-CQA. Lastly, I would like to see an equivalent RNA expression using RT-PCR and a agarose gel figure to visualise the relative transcript abundance in all these treatments. In each of these RT-PCR gel figure you should also provide the amplification of a 'background' or house-keeping RNA to ensure that all conditions are loaded and start with similar amounts of RNA. I see in the methods (section 4.7) you mention RNU6B as an internal control - if this is fulfilling the role of ensuring that all samples contain similar amounts of starting RNA than please provide the gel figure for this as well.
21. Minor method points: which antibiotics were contained in the media in section 4.2? what were the commercial sources of the additives in section 4.2? Wherever optical density or fluorescent measurements are reported in the methods please indicate if the wavelengths being referred to are absorption/excitation/emission signals.
Author Response
Dear Reviewer
Thank you for a constructive and positive review of our manuscript. As you requested, we have enclosed a revised version of manuscript in response to the extensive and insightful reviewer comments. Included below is a point-by-point description of our responses to the reviewer’s comments. We hope that you and the reviewers find this revised manuscript acceptable for publication.
Sincerely
Chau-Jong Wang Ph.D., Professor,
Department of Health Industry Technology Management,
Chung Shan Medical University, Taichung, Taiwan
Tel: (886) 4-24730022, ext. 11670.
E-mail: wcj@csmu.edu.tw
Nov 21, 2021

Reviewer 2 Report
The article entitled “Neochlorogenic acid attenuates hepatic lipid accumulation and inflammation via regulating miR-34a in vitro” is of interest since this compound could be a promising therapy for the treatment of NAFLD. The manuscript is well written, the introduction focuses well on the topic of interest, the objectives are clear and the methodology used is current and in accordance with the results obtained. Authors have employed several in vitro strategies to dilucidate the impact of the treatment with this compound in the synthesis the novo of cholesterol and fatty acids. Moreover they have also studied its effect on inflammatory signalling and in the role of miR-34a targetting SIRT/AKT parhway, which tiggers lipogenesis. Discussion summarizes the main results and compares them with the existing bibliography. The manuscript highlighted that neochlorogenic acid alleviates lipid accumulation by downregulating miR-34a, leading to activation of the SIRT1/AMPK pathway in an in vitro model of NAFLD in HepG2 cells. I would like to recommend this article for publication in the present form
Author Response

(The authors gave the same response as above.)

Round 2
Reviewer 1 Report
A very nice response in general to some concerns with the original manuscript.
I have just a few minor comments and additions that might be made concerning referencing. The research is quite solid.
Minor Points
- I think the author's have misunderstood my criticisms of some of the referencing in the introduction/discussion. In general, I was asking for they themselves to find the appropriate primary research papers for the information they provided, not simply reviews or inapproriate primary research papers as they did. I gave numerous examples of this, but the main point was that the author's should make the effort to find the first (or if several research papers ) several primary research papers that cumulatively support the information being supplied. I had hoped that the author's would make some effort themselves to correct this on a wider scale over the whole manuscript not just the 6-7 examples I had pointed out in the introduction. As a result, the referencing is still, in numerous papers, inappropriate and just credit is not being given to other scientists' papers which have been left out. I'll give a few examples but I would encourage the author's to spend the time checking their information and making sure the correct and appropriate references are supplied. For example:
a) ref 3 and 4 are not the most relevant or the first research papers which illustrated the link between SREO2 and cholesterol synthesis in the liver - a quick pubmed search and literature review canvassing were able to show this. These are several of the key primary research papers which established this point and should be cited: 10.1172/JCI2961, https://doi.org/10.1073/pnas.1534923100, with an excellent review (which should be consulted to see if the author's have got their referencing correct) is http://www.genesdev.org/cgi/doi/10.1101/gad.1854309.
b) ref 5 on SREP2 and AMPK research may summarise this link from other research but it is neither the first nor the most appropriate publication to reference. Again, even as a non-expert I could quickly trace the first and most important papers that made the SREP2-AMPK link, specifically in the contect of lipid accumulation in the liver. These include: doi: 10.1371/journal.pone.0067532, doi: 10.1371/journal.pone.0135637, doi: 10.1016/j.ebiom.2016.02.041, doi: 10.1016/j.nutres.2017.05.01, doi: 10.1096/fj.202001234R, and the original research making the inhibitory link (which should always be cited when the link is made) doi.org/10.1016/j.cmet.2011.03.009
The aim of my comments of referencing was not to keep myself personally happy as a reviewer but to to a a robust and accurate job of reflecting the literature as a background to NAFLD and SREP2 etc regulation in the liver. We spend much time teaching young research students the value and duty they have in citing the literature accurately and comprehensively when they draft papers and it is disappointing to see this is not always followed in otherwise solid research papers. I implore you again: please thoroughly check all the information you have provided in the introduction/methods/ discussion (not just the few further examples I give above) and ensure the references are accurate (the first research demonstrating the information provided) and comprehensive. A research paper of this length should have numerous more references (and more appropriate ones in some instances) than are currently given .
2. I thought I had mentioned this in the comments to the first submission of this paper, and if I have not I apologise, but a paper focused on establishing mechanisms of NAFLD treatment and other closely associated liver metabolic diseases really should provide a brief overview (3-4 sentences, a short paragraph) of the major recent avenues of treatment. This could be done with a simple, brief outline in the introduction, or integrated into the discussion, but in either case, without such an overview your paper lacks some significant research context. To my knowledge, these areas would encompass these examples and references of recent research avenues for treatment:
a) Chloropropanol - DOI: 10.1016/j.lfs.2018.08.007
b) Luteolin - doi: 10.1371/journal.pone.0135637, doi:10.1017/S0007114515001312
c) Metformin - doi: 10.2147/DDDT.S190094, doi: 10.1016/j.freeradbiomed.2018.02.031
d) Endoscopic Bariatric Interventions - recently reviewed in doi: 10.1016/j.cld.2021.08.005
e) GLP-1 Receptor agonists - recently review very well in doi: 10.1155/2021/8936865
f) Treatment of NAFLD and coincident with TIIDM by targeting gut amino acid absorption and amino acid metabolism vis FGF21 signalling, key papers of which are: doi: 10.1111/bph.13711, doi: 10.1016/j.molmet.2015.02.003, 10.1016/j.cmet.2007.05.002
and recently reviewed in: doi: 10.1042/BST20180250, doi: 10.1155/2021/7692447
g) Naringin - doi: 10.1021/acs.jafc.8b02696
h) Hovenia Dulcis - doi: 10.1002/ptr.5741
i) By targeting the gut microbiome - well reviewed in DOI: 10.3389/fmicb.2021.761836
j) Other recent potential pharmacological treatments - well reviewed in doi: 10.21037/tgh-20-247, doi: 10.3748/wjg.v21.i13.3777
- Where the treatment research of NAFLD is very broad and involves much research I have suggested a good review the author's might cite. But in any case, it would be quite easy to reference all the papers listed above and any more key primary ones the author's know of in the field of NAFLD treatment in either a introductory 2/3 sentences or by incorporating them into the current discussion. Please do so.
- In summary, please improve the appropriateness and accuracy of the referencing before the manuscript goes to publication
Author Response
Dear reviewer:
Thank you for a constructive and positive review of our manuscript. As you requested, we have enclosed a revised version of manuscript in response to the extensive and insightful reviewer comments. Included below is a point-by-point description of our responses to the reviewer comments. We hope that you find this revised manuscript acceptable for publication.
Sincerely
Chau-Jong Wang Ph.D., Professor,
Department of Health Industry Technology Management,
Chung Shan Medical University, Taichung, Taiwan
Tel: (886) 4-24730022, ext. 11670.
E-mail: wcj@csmu.edu.tw
Nov 26, 2021
Minor Points
- I think the author's have misunderstood my criticisms of some of the referencing in the introduction/discussion. In general, I was asking for they themselves to find the appropriate primary research papers for the information they provided, not simply reviews or inappropriate primary research papers as they did. I gave numerous examples of this, but the main point was that the author's should make the effort to find the first (or if several research papers ) several primary research papers that cumulatively support the information being supplied. I had hoped that the author's would make some effort themselves to correct this on a wider scale over the whole manuscript not just the 6-7 examples I had pointed out in the introduction. As a result, the referencing is still, in numerous papers, inappropriate and just credit is not being given to other scientists' papers which have been left out. I'll give a few examples but I would encourage the author's to spend the time checking their information and making sure the correct and appropriate references are supplied. For example:
- ref 3 and 4 are not the most relevant or the first research papers which illustrated the link between SREO2 and cholesterol synthesis in the liver - a quick PubMed search and literature review canvassing were able to show this. These are several of the key primary research papers which established this point and should be cited: 10.1172/JCI2961, https://doi.org/10.1073/pnas.1534923100, with an excellent review (which should be consulted to see if the author's have got their referencing correct) is http://www.genesdev.org/cgi/doi/10.1101/gad.1854309.
Answer: Thank you for the precious comment. We have updated and corrected the reference.
- b) ref 5 on SREP2 and AMPK research may summarise this link from other research but it is neither the first nor the most appropriate publication to reference. Again, even as a non-expert I could quickly trace the first and most important papers that made the SREP2-AMPK link, specifically in the contect of lipid accumulation in the liver.
These include: doi: 10.1371/journal.pone.0067532, doi: 10.1371/journal.pone.0135637, doi: 10.1016/j.ebiom.2016.02.041, doi: 10.1016/j.nutres.2017.05.01, doi: 10.1096/fj.202001234R, and the original research making the inhibitory link (which should always be cited when the link is made) doi.org/10.1016/j.cmet.2011.03.009
Answer: Thank you for the precious comment. We have updated and corrected the references.
The aim of my comments of referencing was not to keep myself personally happy as a reviewer but to a robust and accurate job of reflecting the literature as a background to NAFLD and SREP2 etc. regulation in the liver. We spend much time teaching young research students the value and duty they have in citing the literature accurately and comprehensively when they draft papers and it is disappointing to see this is not always followed in otherwise solid research papers. I implore you again: please thoroughly check all the information you have provided in the introduction/methods/ discussion (not just the few further examples I give above) and ensure the references are accurate (the first research demonstrating the information provided) and comprehensive. A research paper of this length should have numerous more references (and more appropriate ones in some instances) than are currently given.
Answer: Thank you for the precious comment. We have updated and checked all the references in the introduction, methods, and discussion.
- I thought I had mentioned this in the comments to the first submission of this paper, and if I have not I apologise, but a paper focused on establishing mechanisms of NAFLD treatment and other closely associated liver metabolic diseases really should provide a brief overview (3-4 sentences, a short paragraph) of the major recent avenues of treatment. This could be done with a simple, brief outline in the introduction, or integrated into the discussion, but in either case, without such an overview your paper lacks some significant research context. To my knowledge, these areas would encompass these examples and references of recent research avenues for treatment:
- a) Chloropropanol - DOI: 10.1016/j.lfs.2018.08.007
- b) Luteolin - doi: 10.1371/journal.pone.0135637, doi:10.1017/S0007114515001312
- c) Metformin - doi: 10.2147/DDDT.S190094, doi: 10.1016/j.freeradbiomed.2018.02.031
- d) Endoscopic Bariatric Interventions - recently reviewed in doi: 10.1016/j.cld.2021.08.005
- e) GLP-1 Receptor agonists - recently review very well in doi: 10.1155/2021/8936865
- f) Treatment of NAFLD and coincident with TIIDM by targeting gut amino acid absorption and amino acid metabolism vis FGF21 signalling, key papers of which are: doi: 10.1111/bph.13711, doi: 10.1016/j.molmet.2015.02.003, 10.1016/j.cmet.2007.05.002
and recently reviewed in: doi: 10.1042/BST20180250, doi: 10.1155/2021/7692447
- g) Naringin - doi: 10.1021/acs.jafc.8b02696
- h) Hovenia Dulcis - doi: 10.1002/ptr.5741
- i) By targeting the gut microbiome - well reviewed in DOI: 10.3389/fmicb.2021.761836
- j) Other recent potential pharmacological treatments - well reviewed in doi: 10.21037/tgh-20-247, doi: 10.3748/wjg.v21.i13.3777
Where the treatment research of NAFLD is very broad and involves much research I have suggested a good review the author's might cite. But in any case, it would be quite easy to reference all the papers listed above and any more key primary ones the author's know of in the field of NAFLD treatment in either an introductory 2/3 sentences or by incorporating them into the current discussion. Please do so.
In summary, please improve the appropriateness and accuracy of the referencing before the manuscript goes to publication
Answer: Thank you for your kind comments. We have mentioned these mechanisms of NAFLD treatment in our discussion section. Please see Page 9.